# Development and Initial Validation of the Assessment of Sleep Environment (ASE): Describing and Quantifying the Impact of Subjective Environmental Factors on Sleep

**DOI:** 10.3390/ijerph192013599

**Published:** 2022-10-20

**Authors:** Michael A. Grandner, Dora Y. Valencia, Azizi A. Seixas, Kayla Oliviér, Rebecca A. Gallagher, William D. S. Killgore, Lauren Hale, Charles Branas, Pamela Alfonso-Miller

**Affiliations:** 1Sleep and Health Research Program, Department of Psychiatry, University of Arizona, Tucson, AZ 85724, USA; 2Department of Population Health, NYU Langone Medical Center, New York, NY 10016, USA; 3Center for Sleep and Circadian Neurobiology, University of Pennsylvania, Philadelphia, PA 19104, USA; 4SCAN Lab, Department of Psychiatry, University of Arizona, Tucson, AZ 85724, USA; 5Department of Family, Population, and Preventive Medicine, Program in Public Health, Stony Brook University, Stony Brook, NY 11794, USA; 6Department of Epidemiology, Columbia University, New York, NY 10032, USA; 7Northumbria Sleep Research, Department of Psychology, Northumbria University, Newcastle upon Tyne NE1 8ST, UK

**Keywords:** sleep, physical environment, sleep quality, insomnia

## Abstract

The purpose of this study was to develop and test the reliability and validity of a 13-item self-report Assessment of Sleep Environment (ASE). This study investigates the relationship between subjective experiences of environmental factors (light, temperature, safety, noise, comfort, humidity, and smell) and sleep-related parameters (insomnia symptoms, sleep quality, daytime sleepiness, and control over sleep). The ASE was developed using an iterative process, including literature searches for item generation, qualitative feedback, and pilot testing. It was psychometrically assessed using data from the Sleep and Healthy Activity Diet Environment and Socialization (SHADES) study (N = 1007 individuals ages 22–60). Reliability was determined with an internal consistency and factor analysis. Validity was evaluated by comparing ASE to questionnaires of insomnia severity, sleep quality, daytime sleepiness, sleep control, perceived stress, and neighborhood disorder. The ASE demonstrated high internal consistency and likely reflects a single factor. ASE score was associated with insomnia symptoms (B = 0.09, *p* < 0.0001), sleep quality (B = 0.07, *p* < 0.0001), and sleep control (B = −0.01, *p* < 0.0001), but not daytime sleepiness. The ASE was also associated with perceived stress (B = 0.20, *p* < 0.0001) and neighborhood disorder (B = −0.01, *p* < 0.0001). Among sleep environment factors, only smell was not associated with sleep quality; warmth and safety were negatively associated with sleepiness; and of the sleep environment factors, only light/dark, noise/quiet, and temperature (warm/cool) were not associated with insomnia symptoms. The ASE is a reliable and valid measure of sleep environment. Physical environment (light, temperature, safety, noise, comfort, humidity, and smell) was associated with insomnia symptoms and sleep quality but not sleepiness.

## 1. Introduction

Sleep health is critical for overall well-being, including cardiometabolic health, mental health, and longevity [1]. Much prior research focuses on how biomedical, demographic, and psychosocial factors may contribute to disturbed sleep. Less research investigates how physical environmental factors, such as bedroom environment (proximal), neighborhood characteristics (distal), and housing conditions, can affect sleep duration, quality, wakefulness, sleep architecture (sleep stages), and circadian rhythms [2].

In particular, environmental factors such as light/illumination, noise, humidity, air temperature, radiant temperature, bedroom air quality, and carbon dioxide concentration affect an individual’s sleep, such as duration and quality [3,4]. Neighborhood characteristics, such as safety, noise levels, neighborhood disadvantage, walkability, social engagement destinations, street intersections, and population density are directly and indirectly linked with a variety of sleep outcomes, such as duration, timing, and quality [2,5,6,7] as well. Housing conditions such as structural environment of home, household crowding, and bed sharing affect sleep duration and quality, as well as sleep disorders, such as insomnia [8,9,10].

Of all the physical environmental factors that affect sleep, light exposure, noise, and temperature are the most widely studied. Exposure to light at night can cause reduced sleep quality and circadian rhythm shifts [11,12]. Noise stimuli can affect quantity and quality of sleep and may induce disruptive arousals during sleep [13,14,15,16]. Temperatures that are too hot or cold can have an adverse impact on sleep, especially during REM sleep [17].

However, noise, light, and temperature do not account for all potential environmental determinants of sleep, and together, they have received relatively little systematic investigation. Recent data highlight that other novel determinants such as neighborhood safety [18] and the sleeping surface affect sleep [17,19]. However, there are no known scales to systematically assess the influence of these environmental factors on sleep.

Traditional measures of environmental determinants of sleep such as the Noise Sensitivity Questionnaire (NoiSeQ [20,21,22]), a measure of an individual’s sensitivity and annoyance to noise during leisure, work, habitation, and communication, is limited in scope as it only measures one factor and emphasizes the psychological reaction to noise, as opposed to determining whether noise and which decibel levels affect an individual’s sleep most. The Pittsburgh Sleep Quality Index (PSQI) [23] contains several individual items on environmental disturbances, but these are nonspecific. The Young Adult Sleep Environment Index (YASEI) measures the sleep environment in general, but it was not designed for generalization to an adult population and it focuses on specific disturbances (e.g., creaking floorboards) rather than domains (e.g., noise) [24].

To incorporate traditional and non-traditional environmental determinants (noise, light, temperature, safety, comfort, humidity, and smell) of sleep, we developed a self-report sleep environment survey that assesses a more comprehensive list of environmental determinants of sleep. We also investigated the psychometric properties (reliability and validity) of our survey and its association with sleep quality, insomnia symptoms, and daytime sleepiness.

## 2. Materials and Methods

### 2.1. Data Source

The Sleep and Healthy Activity, Diet, Environment, and Socialization (SHADES) study, a community-based study of N = 1007 adults ages 22–60 in the Philadelphia area, was used to develop the Assessment of Sleep Environment (ASE). The SHADES study recruited a diverse, community-based sample through advertisements and engagement with community centers. This was not a random sample (e.g., based on random digit dialing), but it also was not a convenience sample (e.g., relying on accessible individuals). Recruitment was focused on community outreach, flyers and advertisements, in-person drives at local community centers, and other efforts. For these reasons, the sample may not be generalizable to the population, but it does represent a diverse group from which to draw inferences. Of note, several other studies have used the SHADES dataset to draw meaningful and useful conclusions regarding a wide range of outcomes [25,26,27,28,29,30]. The original intent of the study was to evaluate cross-sectional relationships between sleep health and behavioral, social, and environmental factors. This study was conducted in 2012–2015. The Institutional Review Boards of the University of Pennsylvania and University of Arizona approved this study, and all study procedures were in accordance with the Declaration of Helsinki.

### 2.2. Scale Development

The Assessment of Sleep Environment (ASE) questionnaire, which assesses the subjective experience of environmental disruptions on sleep, was developed through an iterative process.

Step 1: Surveying existing literature. First, we conducted a review on PubMed and PsycInfo of existing measures that assessed physical environmental determinants of sleep. We performed a content and thematic analysis to determine convergence across the surveys and scales. Then, we performed another review of the literature to determine whether there were additional environmental correlates of sleep that were not captured in the surveys. Our keyword search included the following terms: sleep, circadian, somnolence, sleepiness, external, and environment.

Step 2: Determining original items: Three reviewers (2 primary and a tie-breaker) assessed associations’ parameters in published studies (correlations and likelihood ratios) to determine which environmental factors had the highest association with sleep consistently across the manuscripts.

Step 3: Refining items: The list of environmental factors shown to be associated with sleep was then shown to several experts in the field and non-scientists to determine qualitative feedback on the ecological and construct validity of the list. These individuals suggested additional factors that were not represented. Through this process, a final list of 13 items was generated.

Step 4: Determining answer choices: Once the list of 13 items were generated, we then explored options for an appropriate response scale. Two forms of response categories were qualitatively pilot tested. One included four potential response categories (“Strongly Agree”, “Agree”, “Disagree”, and “Strongly Disagree”) and one included five (the previous response categories, and “Neither Agree nor Disagree” as a neutral option). Since the “Neither Agree nor Disagree” category was rarely chosen during pilot testing and eliminating it affords the ability to determine general “agreement” vs. “disagreement,” it was eliminated from the item list on the final survey.

### 2.3. Finalizing the ASE Scale

The final instrument assesses the degree to which sleep was associated with environment, such as being too bright or dark, too noisy or quiet, too warm or cool, or too humid; having an uncomfortable smell; being uncomfortable because of pillows or blankets; being too firm or too soft; being uncomfortable for another reason; or not feeling safe.

The sole prompt for the questionnaire is “I have difficulty sleeping because the place where I sleep…”; this was chosen at the outset of the measure development and remained throughout. No time criterion was provided, so that this can be left up to the interpretation of the participant to reflect general experience. Since environmental issues may have variable frequencies of presentation, this allows for the ability to account for this. Furthermore, the questionnaire does not include a time-of-day specifier because not all people sleep at night. Additionally, no location specifier was included because not all people sleep in a bed. The prompt was worded to be as general as possible. All items are presented with 4 possible response categories: “Strongly Agree”, “Agree”, “Disagree”, or “Strongly Disagree”. The questionnaire was scored by assigning values of 0–3 for each item (0 = strongly disagree and 3 = strongly agree) and summing the values.

### 2.4. Assessment of Reliability

To assess the reliability of the ASE, several approaches were undertaken. Internal consistency was evaluated with several approaches: (1) spearman correlations among all pairs of items, (2) Cronbach’s alpha, (3) split-half correlation, and (4) factor analysis.

### 2.5. Assessment of Validity

To assess validity, the ASE score was evaluated relative to several outcomes of interest in order to determine whether they are related to overall environmental disturbance as hypothesized. Convergent validity was assessed relative to measures of overall sleep quality, insomnia severity, daytime sleepiness, and perceived control over sleep. In addition, convergent validity was explored relative to additional factors hypothesized to be related to environmental disturbances to sleep, including daytime stress and neighborhood quality.

Overall sleep quality was assessed with the Pittsburgh Sleep Quality Index (PSQI) [23]. This is a widely used screening tool to identify individuals with “good” versus “poor” sleep quality among 7 components: sleep duration, perceived sleep quality, sleep latency, sleep efficiency, sleep disturbance, use of sleep aids, and daytime disturbance. It was hypothesized that ASE scores would be positively associated with worse sleep quality (higher PSQI scores).

Insomnia severity was assessed using the Insomnia Severity Index (ISI) [31]. The ISI is a standard instrument for use in insomnia research and is used to identify diagnostic risk and track treatment progression in insomnia [32]. This questionnaire assesses the degree to which insomnia symptoms cause concern or impairment. It was hypothesized that higher ASE scores would be associated with worse insomnia severity (higher ISI scores).

Daytime sleepiness was assessed with the Epworth Sleepiness Scale (ESS) [33,34,35]. The ESS is a standard assessment of habitual daytime sleepiness and rates the degree to which an individual is likely to fall asleep or doze in a variety of different situations. High scores indicate increased daytime sleepiness. It was hypothesized that higher ASE scores would be associated with greater daytime sleepiness (higher ESS scores).

The degree to which an individual has control over their sleep was assessed using the Brief Index of Sleep Control (BRISC) [25]. The BRISC is a 4-item measure that assesses the degree to which an individual believes that they have control over when they go to bed, when they get up, how much they sleep, and how well they sleep. It was hypothesized that ASE scores would be negatively associated with BRISC scores (indicating less perceived control over sleep).

Daytime stress was assessed with the Perceived Stress Scale (PSS) [36]. The PSS is a standard measure of general perceived stress, across a wide range of situations. It was hypothesized that higher ASE scores would be associated with higher PSS scores (greater perceived stress).

Neighborhood quality was associated with the Neighborhood Disorder Scale [37]. This scale measures neighborhoods along 2 dimensions—order and disorder—and results in a total score representing the ratio of order to disorder (higher scores indicate more neighborhood disorder). It was hypothesized that living in a more disordered neighborhood would be associated with a higher ASE score.

### 2.6. Covariates

Covariates included age (computed based on self-reported date of birth), sex (self-reported), income (total household income divided by quintile), and subjective social status (SSS). SSS was assessed using the “SSS ladder [38]” where individuals rated their status on a scale of 1–10 relative to others in their community and relative to the nation in general.

### 2.7. Statistical Analyses

To assess convergent validity, we performed several regression analyses to investigate the relationship between ASE (predictor/independent variable) and the following outcomes: PSQI, ISI, ESS, BRISC, PSS, and neighborhood scores, while adjusting for age and sex. Post hoc analyses examined relationships between each ASE item and the aforementioned outcomes. These secondary analyses may provide insight regarding which environmental factors might be related to the certain outcomes of interest. All statistical analyses were performed using STATA 14.0 (STATA Corp, College Station, TX, USA).

## 3. Results

### 3.1. Characteristics of the Sample

The results were compiled using 1007 individuals from the Sleep and Healthy Activity, Diet, Environment, and Socialization (SHADES) study, ages 22–60 in the Philadelphia area. More complete characteristics of this sample have been reported elsewhere. Distributions of variables used in the present analysis are reported in Table 1. Overall, the mean age of the sample was 34 (SD = 9.4 years), and the sample was 61% female.

Overall mean values for insomnia severity, sleepiness, and sleep quality are also reported in Table 1, with mean scores of 10.6, 7.8, and 8.3 on the ISI, ESS, and PSQI, respectively. The mean score on the ASE was 23.9 (SD = 9.3). In univariate analyses, ASE scores did not differ by age (B = 0.01, *p* = 0.687) or sex, (B = −1.03, *p* = 0.09). Table 1 also reports the distribution of responses to all of the ASE items, indicating that there were adequate cell sizes for evaluating by category in secondary analyses. The Cronbach’s alpha for the PSQI, ISI, ESS, BRISC, PSS, and neighborhood disorder assessment were 0.77 (based on component scores), 0.89, 0.80, 0.80, 0.86, and 0.93, respectively.

### 3.2. Reliability

The ASE demonstrated high internal consistency (Cronbach’s alpha = 0.9); see Table 2 for correlations among individual items. All individual ASE items correlated with each other at the 0.0005 level. In addition, the ASE demonstrated a split-half correlation of r = 0.8, *p* ≤ 0.0005. A principal components exploratory factor analysis was conducted to assess construct reliability among the items and a single-factor solution (1 eigenvalue > 1.0) was found, with a Factor 1 eigenvalue of 5.18, representing 93.3% of the variance, with a chi-square test of independence (df = 78) = 5239.3, *p* < 0.0005. Factor loadings were as follows: light (0.56), dark (0.58), noise (0.57), quiet (0.60), warm (0.55), cool (0.56), humid (0.64), smell (0.69), pillow (0.68), firm (0.74), soft (0.69), other surface issue (0.72), and safety (0.61).

### 3.3. Validity

Relationships between ASE and PSQI, ISI, ESS, and BRISC scores are reported in Table 3. Overall, higher ASE scores were associated with higher PSQI scores, indicating worse sleep quality (B = 0.07, *p* < 0.0001). Higher ASE scores were also associated with higher ISI scores, indicating greater insomnia severity (B = 0.09, *p* < 0.0001). ASE scores were not associated with ESS scores, indicating no statistically significant association with daytime sleepiness. ASE scores were negatively associated with BRISC scores, indicating less control of one’s sleep (B = −0.01, *p* < 0.0001). Post hoc analyses of individual items are also reported in Table 3. Given a Bonferroni correction to the significance level of 0.05/13 items = 0.0038, several relationships between individual items and sleep-related variables were found. Using this criterion, higher PSQI scores were associated with higher ratings of items on the ASE reflecting that sleep quality was disturbed by an environment that was too dark, too quiet, too warm, too cool, or too humid; had an uncomfortable pillow; was too firm; had some other problem with the sleeping surface; and felt unsafe. Higher ISI scores were associated with environments that were too humid, had uncomfortable pillows, was too firm or too soft, had some other issue with the sleeping surface, or was not safe. Higher BRISC scores were negatively associated with a sleep environment that was too humid, has an uncomfortable pillow, had some other sleeping surface issue, and was not safe. In models that included additional adjustment for socioeconomic factors, these relationships were generally maintained. Total ASE score was still associated with higher PSQI (B = 0.06, *p* < 0.0001), ISI (B = 0.07, *p* < 0.0001), and BRISC (B = −0.01, *p* = 0.001) scores, but not ESS. Relationships with individual items were also generally maintained (see Table 3).

Associations between ASE scores and perceived stress (assessed with the PSS) and neighborhood disorder are reported in Table 4. Higher ASE scores were associated with higher levels of perceived stress (B = 0.2, *p* < 0.0001) and neighborhood disorder (B = 0.01, *p* < 0.0001). Regarding post hoc analyses of individual items, using a Bonferroni-corrected cutoff of *p* < 0.0038, greater perceived stress was associated with greater endorsement of environments that were too dark, were too noisy, were too warm or cool, were too humid, had an uncomfortable smell, had uncomfortable pillows, were too firm, were too soft, had some other issue with the sleeping surface, and were not safe. Increased neighborhood disorder was associated with sleep environments that were too dark, too noisy, too humid, too firm, too soft, and not safe. In models that also included socioeconomic factors (also in Table 3), associations were generally maintained (coefficients not reduced). Higher ASE scores were also associated with greater levels of stress (B = 0.17, *p* < 0.0001) and neighborhood disorder (B = 0.01, *p* < 0.0001), and most relationships with individual items remained consistent.

ASE scores were then categorized as low (0–9), moderate (10–19), and high (20 or higher). Table 5 displays the means and standard deviations of PSQI, ISI, ESS, BRISC, PSS, and neighborhood disorder scores at each of these three levels of ASE scores. In addition, it shows that scores on these questionnaires were statistically significantly higher in the higher ASE categories. Specifically, compared to those with low ASE scores, those with moderate scores demonstrated, on average, 1.3 more points on the PSQI (*p* < 0.0001), 1.7 more points on the ISI (*p* = 0.001), 0.25 fewer points on the BRISC (*p* < 0.0001), 3.3 more points on the PSS (*p* < 0.0001), and 0.16 points higher on the neighborhood disorder screener (*p* < 0.0001). Those with high scores reported on average 1.9 more points on the PSQI (*p* < 0.0001), 2.5 more points on the ISI (*p* < 0.0001), 0.32 fewer points on the BRISC (*p* < 0.0001), 4.8 more points on the PSS (*p* < 0.0001), and 0.19 more points on the neighborhood disorder screener (*p* < 0.0001).

## 4. Discussion

The present study describes the development and initial validation of the Assessment of Sleep Environment (ASE). The ASE assesses several environmental domains that may impact sleep. Overall, the ASE has strong internal consistency and was associated with insomnia and poor sleep quality. The ASE had adequate validity with global insomnia and sleep quality, but not with daytime sleepiness. Based on our findings, ASE was associated with insomnia, sleep quality, and daytime sleepiness.

### 4.1. The Novelty and Utility of the ASE

To our knowledge, the Assessment of Sleep Environment (ASE) is the first comprehensive instrument that captures and operationalizes multiple environmental factors that may affect sleep. First, unlike the ASE, previously developed measures such as the Noise Sensitivity Questionnaire (NoiSeQ), a measure of an individual’s sensitivity and annoyance to noise during leisure, work, habitation, communication, and sleep, only measures one environmental stimuli—noise, thus limiting its ability to capture a wide range of environmental factors that affect sleep. Second, previously developed measures such as the NoiSeQ have unsuccessfully measured the impact of noise on several sleep disturbance parameters, such as insomnia and daytime sleepiness [20,21,22]. Third, the NoiSeQ is limited in scope because it assumes and posits a putative association between noise exposure, annoyance, and sleep disturbance, where noise exposure causes annoyance and annoyance causes sleep disturbance. The relationship between annoyance and sleep disturbance may be mediated by habituation to noise levels, psychological factors (e.g., personality, coping and temperament), and how sleep deprived the individual is which can decrease their threshold of getting annoyed by noise levels. The NoiSeQ, like previously developed measures, focuses on the psychological reaction to environmental stimuli and makes an inference as to whether that stimuli affects sleep.

Conversely, the ASE asks directly whether the individual believes a certain environmental stimulus affects their sleep. The advantage of the ASE is that it can be used for several purposes across diverse settings. For example, the ASE could be used in a medical setting, specifically inpatient settings, to assess how a healthcare facility’s environment affects patients’ sleep. It could presumably be utilized as an independent variable in studies examining potential influences on sleep (e.g., to what degree the sleep environment impacts outcomes of interest), as a mediator or moderator variable in studies examining sleep interventions (e.g., to what degree does a healthy sleep environment impact the effectiveness of an intervention), or as an outcome variable (e.g., does a noise reduction effort result in less environmental sleep disturbance). Future studies will be able to explore the ASE in the context of more diverse populations in terms of race/ethnicity, geographic location, nationality, and acculturation, amongst others. In spite of the advantages, the ASE is limited in its reliance on individual insight about the relationship between environmental stimuli and their sleep, a notion known as causal inference bias.

It should be noted that the ASE separately assesses seemingly orthogonal constructs such as an environment being both “too noisy” and/or “too quiet.” At first glance, it may not be clear how an environment can be both too noisy and too quiet, and how more “noisiness” is not necessarily indicative of less “quietness.” The ASE keeps these items (and other dichotomies in the measure) separate for three reasons. First, it is possible for loudness and quietness, light and dark, or other dichotomies to exist together. For example, perhaps an environment is disturbingly quiet most of the time but is sometimes punctuated by loud traffic noises. Or perhaps individuals have difficulty sleeping because it is pitch black, which causes anxiety, but their bedpartner wakes in the middle of the night and turns on lights. There are many such possible scenarios where these can co-occur, or at least exist on independent spectra. Second, the scale does not measure how noisy or quiet the environment actually is. Rather, the scale measures the degree to which each element was specifically disruptive. Although there is likely a correlation between actual noise level and the degree to which one might be expected to have their sleep disturbed by noise, these are independent constructs. Third, the mechanisms linking each item to sleep disturbance may be variable. For example, an environment may disturb sleep by being “too quiet” because it triggers fears and anxiety or rumination, but the link with being “too noisy” may have more to do with sensory gating and arousal thresholds. Therefore, since the possible mechanisms are different, they are evaluated separately.

### 4.2. Reliability and Validity of ASE in Sleep Domains

The ASE demonstrated good reliability and adequate validity. First, the two reliability properties, internal consistency and construct reliability, indicate that all 13 items in the ASE measure sleep environment reliably, a sign of a well-developed measure. The high split-half reliability implies that all 13 items in the ASE are equally important and undercuts previous research emphasis on (e.g., noise, temperature, and light) and omission of (safety, mattress, and pillow features) certain environmental factors. Our inclusion of non-traditional environmental factors such as safety, mattress and pillow features are supported by previous research showing an association with sleep. For example, Chen and colleagues found that mattresses that regulate distribution of pressure were associated with better sleep quality [39]. This is supported by work by McCall and colleagues that showed that pressure distribution on home mattresses was associated with sleep quality [40]. In addition to pressure distribution, other aspects seem to be relevant. Verhaert and colleagues showed that beds with ergonomics that provided better spinal alignment were associated with better sleep quality [41]. Shanmugan and colleagues found that firmer mattresses might be better for pain alleviation [42].

### 4.3. Associations between Sleep Environment and Sleep Parameters

Findings from our study indicated that sleep environment was associated with greater insomnia symptoms, poorer sleep quality, and less control of one’s sleep, but not daytime sleepiness. Insomnia and sleep environment were significantly associated, and certain environmental factors (light, dark, humid, pillow, firm, soft, other surface, and safety) had very pronounced associations with insomnia symptoms. These findings indicate individuals with higher levels of insomnia are more likely to identify a wide selection of environmental factors that affect their sleep. Sleep quality was associated with the following sleep environment factors: darkness of room, how noisy was the sleep environment, warmth of the sleep environment, coolness/temperature, humid, pillow, mattress (firm or soft), other surface such as mattress, and how safe the individual felt while asleep, had pronounced associations with sleep quality. Less control of sleep was associated with ASE scores suggesting that disturbances in sleep caused by sleep environment affected individual’s perception of having control of their sleep such as when they go to bed, when they awake, how much sleep they get, and how well they sleep. It is not necessarily the case that that these environmental factors caused insomnia, poor sleep quality or less control over sleep, but it is likely they directly or indirectly affect these outcomes. We argue that the reason why daytime sleepiness was not significantly associated with global sleep environment is because daytime sleepiness is independent of sleep environment and thus may be mediated or moderated by other factors that are more proximally related, which may mask putative associations between sleep environment and daytime sleepiness. Unlike insomnia, sleep quality, and control over sleep, it is likely that the association between daytime sleepiness and sleep environment is mediated by other factors, such as poor sleep and sleep deprivation.

There are a number of physiologic pathways that could explain these relationships. For example, there is extensive research on environmental light at night as an impediment to sleep health [43,44]. This largely functions as a circadian input, delaying sleep in the early part of the night and advancing it toward the end of the night because of the stimulation at the retina of photosensitive cells that promote a “day” signal in the suprachiasmatic nucleus. Reasons why darkness may be an impediment to sleep may implicate hyperarousal pathways, whereby some individuals may become alerted by the stress associated with the inability to assess the environment [45]. Noise has been independently investigated in a number of studies as an impediment to sleep health. Several previous studies show that auditory stimulation at night can increase sleep stage transitions, arousals, and awakenings [16,46,47,48]. Quiet, though, may also be problematic. Similar to the situation of darkness, excessive quiet may be unsettling for some people or may lead to lowered arousal thresholds. Temperature is also well-studied in relation to sleep. Many previous studies have documented the relationship between 24 h temperature rhythms and sleep–wake control [49,50,51,52]. For these reasons, other studies have shown that being too hot or cold can adversely impact sleep health [17,53]. Although there is little data on humidity, one previous study found a relationship between habitual sleep health and humidity levels [54]. There is also some research linking olfaction to the sleep experience [55], though this is more focused on likely conditioned responses to smells. Feelings of a lack of safety have also been relatively extensively assessed as promoting hypervigilance and reducing sleep depth [1,18,56]. The mechanisms linking experience of mattresses, pillows, and other sleeping surfaces to sleep experiences are surprisingly understudied. Some research shows that firmness is associated with pain experiences [39,40,41], but despite the fact that most people have a sleeping surface that they habitually use and that there are a large number of claims made about the relative value of these sleeping surfaces for promoting sleep health, more research is needed to understand these mechanisms.

### 4.4. Associations between Sleep Environment and Stress and Neighborhood Characteristics

Our findings also indicate that sleep environment was associated with perceived stress and neighborhood disorder, a concept that characterizes activities, behaviors, and neighborhood structural conditions indicate any breakdown of social control [37]. Although stress and neighborhood disorder are two distinguishable factors, we conjecture their relationship with sleep environment may operate similarly and thus may be explained through two pathways.

The first pathway suggests that stress and neighborhood disorder may be contiguous to sleep environment. Stress, which has been shown to adversely impact sleep duration and quality [57,58], can also affect perceptions of the sleep environment. For example, researchers have found that pre-sleep stress was related to sleep-onset insomnia [59]. In 2004, Tang and Harvey also found that both pre-sleep cognitive and physiological arousal were associated with distorted perceptions about sleep [60]. Difficulty initiating sleep as a result of stress may influence how an individual perceives their immediate sleep environment, as they are more likely to characterize it as negative or not conducive to sleep. Our argument is grounded in previous work that showed that stress, especially chronic stress, activated the stress system in the body via the hypothalamic–pituitary–adrenal axis and cognition [61,62]. It is also likely that the sleep environment may increase an individual’s stress. An environment not conducive to sleep may increase anxiety about initiating and maintaining sleep and thus may increase their stress level [63].

The second argument suggests that neighborhood disorder and stress are inter-related. Neighborhood disorder may operate like a stressor to an individual, thus negatively affecting how they perceive their sleep environment as one that is not conducive to sleep. Previous work showing that neighborhood disorder is linked to stress outcomes, both psychological and physiological, further buttress our argument that it operates like a stressor [64,65].

### 4.5. Limitations

There are methodological (sampling bias, design, and instrumentation) and conceptual limitations with this study and the ASE questionnaire which might potentially affect our findings. First, data for our study relied on cross-sectional design, and as such, we were unable to determine if components of the ASE were reliably measured over time or to assess the long-term effects of sleep environment on sleep disturbance parameters. Cross-sectional associations may result in prevalence-incidence bias. Second, our study relied on self-reported, subjective measures and not objective sleep environment data and thus might be subject to recall bias. In other words, the ASE items have not been validated against objective environmental recordings in sleeping areas. The PSQI is included in analyses, even though it includes an item on the environment being too hot or cold. We did not remove this item from analysis for two reasons; first, the scale is not validated without that item, and second, when we performed a post hoc evaluation, removing the item did not change any of the observed relationships. In addition, rigorous analytic techniques such as item response theory approaches, may further clarify and aid in the refinement of measures such as these. Future work could apply these principles to this and other sleep-related assessments. Despite the foregoing limitations, our findings still have scientific and practical value, as it adds to a growing body of research that investigates the relationship between several sleep environmental factors and sleep disturbance parameters.

## 5. Conclusions

Sleep is an important domain of health and social functioning; however, no standardized instruments exist to assess external environmental factors that impact sleep. The ASE was developed as a solution for this problem, which had demonstrable reliability and adequate validity regarding relationships between subjective experiences of the sleep environment and insomnia, sleep quality, sleep control, and other factors. Future studies should examine the construct of subjective sleep environment as assessed in the ASE to explore how it is differentially experienced by individuals across age, racial/ethnic, socioeconomic, and other societal groups. Additionally, future studies should explore how ASE responses relate to issues such as sleep hygiene, activities in bed, circadian factors such as chronotype and seasonality, and disruptions caused by bedpartners. Future studies should also continue to investigate the utility of the ASE across diverse settings and subgroups to solidify its utility as a potential clinical and public health tool.

## Figures and Tables

**Table 1 ijerph-19-13599-t001:** Characteristics of the Sample.

Variable (Mean, Standard Deviation)	Category/Units	Values
Age	Years	34.0 ± 9.4
Sex	Male	38.53%
	Female	61.47%
Insomnia Severity	ISI Score	10.6 ± 6.3
Daytime Sleepiness	ESS Score	7.84 ± 4.65
Sleep Quality	PSQI Score	8.26 ± 4.18
Sleep Environment	ASE Score	23.9 ± 9.3
Light (1.25, 1.10)	Strongly Disagree	19.66%
	Disagree	16.68%
	Agree	32.17%
	Strongly Agree	31.48%
Dark (0.71, 0.98)	Strongly Disagree	11.92%
	Disagree	2.28%
	Agree	30.88%
	Strongly Agree	54.92%
Noise (1.36, 1.07)	Strongly Disagree	19.46%
	Disagree	23.44%
	Agree	30.49%
	Strongly Agree	26.61%
Quiet (0.85, 1.02)	Strongly Disagree	13.41%
	Disagree	4.97%
	Agree	34.76%
	Strongly Agree	46.87%
Warm (1.64, 1.09)	Strongly Disagree	27.90%
	Disagree	28.40%
	Agree	23.93%
	Strongly Agree	19.76%
Cool (1.31, 1.12)	Strongly Disagree	22.74%
	Disagree	14.80%
	Agree	33.57%
	Strongly Agree	28.90%
Humid (1.43, 1.13)	Strongly Disagree	25.32%
	Disagree	19.36%
	Agree	28.80%
	Strongly Agree	26.51%
Smell (0.82, 1.01)	Strongly Disagree	11.92%
	Disagree	7.35%
	Agree	31.58%
	Strongly Agree	49.16%
Pillow (1.20, 1.10)	Strongly Disagree	18.27%
	Disagree	17.78%
	Agree	29.49%
	Strongly Agree	34.46%
Firm (1.15, 1.11)	Strongly Disagree	19.17%
	Disagree	12.21%
	Agree	32.87%
	Strongly Agree	35.75%
Soft (1.08, 1.08)	Strongly Disagree	17.78%
	Disagree	9.24%
	Agree	35.85%
	Strongly Agree	37.14%
Other Surface (1.30, 1.13)	Strongly Disagree	21.55%
	Disagree	18.67%
	Agree	28.30%
	Strongly Agree	31.48%
Safe (0.96, 1.07)	Strongly Disagree	15.00%
	Disagree	9.93%
	Agree	30.69%
	Strongly Agree	44.39%
Income	Quintile 1	18.6%
	Quintile 2	15.6%
	Quintile 3	24.7%
	Quintile 4	22.3%
	Quintile 5	18.2%
Subjective Social Status	Community	5.27 ± 2.06
	General	5.48 ± 1.98

**Table 2 ijerph-19-13599-t002:** Correlations Among ASE Items (all *p* < 0.0005).

	Total Score	Light	Dark	Noise	Quiet	Warm	Cool	Humid	Smell	Pillow	Firm	Soft	Other Surface
Light	0.617	1.000											
Dark	0.617	0.387	1.000										
Noise	0.623	0.452	0.321	1.000									
Quiet	0.635	0.353	0.536	0.362	1.000								
Warm	0.600	0.330	0.280	0.382	0.316	1.000							
Cool	0.612	0.325	0.346	0.335	0.352	0.444	1.000						
Humid	0.682	0.373	0.336	0.389	0.348	0.554	0.461	1.000					
Smell	0.710	0.382	0.386	0.395	0.431	0.341	0.367	0.434	1.000				
Pillow	0.698	0.363	0.316	0.332	0.305	0.310	0.251	0.379	0.532	1.000			
Firm	0.741	0.357	0.379	0.361	0.387	0.330	0.331	0.420	0.488	0.573	1.000		
Soft	0.701	0.331	0.379	0.315	0.415	0.266	0.334	0.365	0.454	0.530	0.611	1.000	
Other Surface	0.727	0.358	0.350	0.354	0.359	0.326	0.335	0.383	0.439	0.553	0.690	0.624	1.000
Safe	0.649	0.295	0.376	0.370	0.364	0.264	0.296	0.399	0.498	0.443	0.431	0.409	0.461

**Table 3 ijerph-19-13599-t003:** Relationships between ASE Items and Sleep Quality, Insomnia Severity, Daytime Sleepiness, and Sleep Control.

	PSQI Score			ISI Score			ESS Score			BRISC Score		
	B	95% CI	*p*	B	95% CI	*p*	B	95% CI	*p*	B	95% CI	*p*
**Adjusted for Age and Sex**												
Total	0.071	(0.044, 0.098)	<0.0001	0.090	(0.049, 0.132)	<0.0001	0.021	(−0.010, 0.052)	0.189	−0.012	(−0.018, −0.006)	<0.0001
Light	0.115	(−0.117, 0.347)	0.329	0.154	(−0.200, 0.508)	0.393	−0.097	(−0.359, 0.164)	0.464	−0.005	(−0.057, 0.047)	0.858
Dark	0.494	(0.234, 0.754)	<0.0001	0.411	(0.012, 0.809)	0.043	0.352	(0.058, 0.645)	0.019	−0.056	(−0.115, 0.002)	0.059
Noisy	0.306	(0.068, 0.544)	0.012	0.336	(−0.028, 0.699)	0.070	0.017	(−0.252, 0.285)	0.902	−0.036	(−0.089, 0.018)	0.192
Quiet	0.388	(0.138, 0.639)	0.002	0.282	(−0.102, 0.666)	0.149	0.178	(−0.105, 0.461)	0.217	−0.017	(−0.074, 0.039)	0.544
Warm	0.442	(0.208, 0.676)	<0.0001	0.438	(0.080, 0.797)	0.016	0.092	(−0.172, 0.357)	0.494	−0.072	(−0.125, −0.020)	0.007
Cool	0.340	(0.112, 0.567)	0.003	0.299	(−0.049, 0.647)	0.092	0.139	(−0.118, 0.396)	0.289	−0.064	(−0.115, −0.013)	0.013
Humid	0.550	(0.327, 0.772)	<0.0001	0.717	(0.376, 1.058)	<0.0001	0.271	(0.018, 0.524)	0.036	−0.091	(−0.141, −0.041)	<0.0001
Smell	0.173	(−0.081, 0.427)	0.182	0.156	(−0.232, 0.543)	0.431	−0.093	(−0.379, 0.193)	0.522	−0.053	(−0.110, 0.004)	0.068
Pillow	0.473	(0.244, 0.703)	<0.0001	0.783	(0.433, 1.133)	<0.0001	0.082	(−0.179, 0.343)	0.537	−0.107	(−0.158, −0.055)	<0.0001
Firm	0.392	(0.162, 0.622)	0.001	0.582	(0.231, 0.933)	0.001	0.116	(−0.145, 0.376)	0.383	−0.069	(−0.120, −0.017)	0.009
Soft	0.351	(0.116, 0.587)	0.004	0.643	(0.284, 1.002)	<0.0001	0.159	(−0.107, 0.425)	0.242	−0.066	(−0.119, −0.013)	0.014
Other Surface	0.535	(0.311, 0.760)	<0.0001	0.853	(0.510, 1.196)	<0.0001	0.119	(−0.137, 0.375)	0.362	−0.115	(−0.165, −0.065)	<0.0001
Safe	0.695	(0.461, 0.930)	<0.0001	0.927	(0.568, 1.287)	<0.0001	0.221	(−0.047, 0.489)	0.107	−0.100	(−0.153, −0.047)	<0.0001
**Adjusted for Age, Sex, Income, and Subjective Social Status (Overall and Relative to Community)**												
Total	0.061	(0.034, 0.087)	<0.0001	0.074	(0.034, 0.114)	<0.0001	0.013	(−0.018, 0.044)	0.408	−0.010	(−0.016, −0.004)	0.001
Light	0.134	(−0.087, 0.356)	0.235	0.181	(−0.158, 0.519)	0.295	−0.083	(−0.340, 0.174)	0.527	−0.008	(−0.058, 0.043)	0.768
Dark	0.446	(0.197, 0.695)	<0.0001	0.331	(−0.051, 0.713)	0.090	0.315	(0.025, 0.606)	0.033	−0.047	(−0.104, 0.011)	0.111
Noisy	0.282	(0.055, 0.510)	0.015	0.297	(−0.051, 0.644)	0.094	0.002	(−0.262, 0.267)	0.986	−0.030	(−0.082, 0.022)	0.254
Quiet	0.383	(0.144, 0.622)	0.002	0.273	(−0.093, 0.640)	0.144	0.175	(−0.104, 0.454)	0.218	−0.016	(−0.071, 0.039)	0.569
Warm	0.414	(0.191, 0.637)	<0.0001	0.394	(0.052, 0.736)	0.024	0.076	(−0.185, 0.337)	0.569	−0.067	(−0.119, −0.016)	0.010
Cool	0.282	(0.063, 0.500)	0.011	0.205	(−0.129, 0.539)	0.230	0.087	(−0.167, 0.341)	0.502	−0.055	(−0.105, −0.005)	0.030
Humid	0.459	(0.245, 0.673)	<0.0001	0.578	(0.250, 0.906)	0.001	0.208	(−0.043, 0.458)	0.104	−0.076	(−0.125, −0.027)	0.002
Smell	0.137	(−0.106, 0.380)	0.270	0.098	(−0.274, 0.469)	0.606	−0.123	(−0.405, 0.160)	0.394	−0.044	(−0.100, 0.011)	0.118
Pillow	0.402	(0.182, 0.623)	<0.0001	0.673	(0.336, 1.009)	<0.0001	0.027	(−0.231, 0.285)	0.839	−0.094	(−0.144, −0.043)	<0.0001
Firm	0.292	(0.070, 0.513)	0.010	0.423	(0.085, 0.762)	0.014	0.045	(−0.213, 0.303)	0.733	−0.051	(−0.102, 0.000)	0.050
Soft	0.294	(0.067, 0.520)	0.011	0.548	(0.204, 0.893)	0.002	0.112	(−0.151, 0.376)	0.404	−0.056	(−0.108, −0.004)	0.033
Other Surface	0.395	(0.177, 0.613)	<0.0001	0.641	(0.309, 0.974)	<0.0001	0.012	(−0.243, 0.267)	0.927	−0.093	(−0.143, −0.043)	<0.0001
Safe	0.579	(0.352, 0.807)	<0.0001	0.747	(0.399, 1.095)	<0.0001	0.136	(−0.131, 0.403)	0.317	−0.078	(−0.131, −0.026)	0.003

**Table 4 ijerph-19-13599-t004:** Relationships between ASE Items and Related Factors of Perceived Stress and Neighborhood Disorder.

	Perceived Stress Scale			Neighborhood Disorder		
	B	95% CI	*p*	B	95% CI	*p*
**Adjusted for Age and Sex**						
Total Score	0.195	(0.140, 0.251)	<0.0001	0.007	(0.004, 0.010)	<0.0001
Light	0.634	(0.158, 1.110)	0.009	0.030	(0.003, 0.058)	0.032
Dark	0.957	(0.422, 1.492)	<0.0001	0.055	(0.024, 0.086)	0.001
Noisy	0.826	(0.337, 1.314)	0.001	0.064	(0.035, 0.092)	<0.0001
Quiet	0.606	(0.090, 1.123)	0.021	0.020	(−0.010, 0.050)	0.184
Warm	1.106	(0.626, 1.585)	<0.0001	0.032	(0.004, 0.060)	0.026
Cool	1.103	(0.638, 1.568)	<0.0001	0.037	(0.010, 0.065)	0.007
Humid	1.211	(0.754, 1.669)	<0.0001	0.046	(0.019, 0.072)	0.001
Smell	1.085	(0.566, 1.604)	<0.0001	0.034	(0.004, 0.064)	0.029
Pillow	1.205	(0.734, 1.676)	<0.0001	0.033	(0.006, 0.061)	0.018
Firm	1.286	(0.817, 1.756)	<0.0001	0.052	(0.024, 0.079)	<0.0001
Soft	1.037	(0.554, 1.520)	<0.0001	0.045	(0.017, 0.073)	0.002
Other Surface	1.701	(1.246, 2.157)	<0.0001	0.040	(0.013, 0.067)	0.004
Safe	1.631	(1.151, 2.112)	<0.0001	0.047	(0.018, 0.075)	0.001
**Adjusted for Age, Sex, Income, and Subjective Social Status (Overall and Relative to Community)**						
Total Score	0.171	(0.120, 0.222)	<0.0001	0.007	(0.003, 0.010)	<0.0001
Light	0.686	(0.247, 1.124)	0.002	0.030	(0.003, 0.058)	0.032
Dark	0.876	(0.381, 1.371)	0.001	0.051	(0.020, 0.082)	0.001
Noisy	0.780	(0.329, 1.230)	0.001	0.061	(0.033, 0.090)	<0.0001
Quiet	0.620	(0.144, 1.096)	0.011	0.018	(−0.012, 0.048)	0.233
Warm	1.012	(0.570, 1.454)	<0.0001	0.033	(0.005, 0.061)	0.021
Cool	0.952	(0.520, 1.383)	<0.0001	0.036	(0.009, 0.063)	0.009
Humid	0.992	(0.567, 1.417)	<0.0001	0.042	(0.015, 0.069)	0.002
Smell	1.022	(0.543, 1.502)	<0.0001	0.031	(0.001, 0.061)	0.045
Pillow	1.055	(0.618, 1.492)	<0.0001	0.029	(0.001, 0.056)	0.042
Firm	1.049	(0.612, 1.486)	<0.0001	0.047	(0.019, 0.074)	0.001
Soft	0.919	(0.472, 1.367)	<0.0001	0.041	(0.013, 0.069)	0.005
Other Surface	1.334	(0.906, 1.762)	<0.0001	0.038	(0.011, 0.065)	0.007
Safe	1.357	(0.907, 1.806)	<0.0001	0.041	(0.013, 0.070)	0.005

**Table 5 ijerph-19-13599-t005:** ASE Score Categories Associated with Sleep Quality, Insomnia Severity, Daytime Sleepiness, Sleep Control, Perceived Stress, and Neighborhood Disorder.

		M (SD)	B	95% CI	*p*
PSQI	Low (0–9)	7.1 (4.3)	1.000	Reference	
	Moderate (10–19)	8.4 (4.1)	1.283	(0.671, 1.895)	<0.0001
	High (20–39)	9.0 (4.0)	1.850	(1.189, 2.510)	<0.0001
ISI	Low (0–9)	9.1 (6.7)	1.000	Reference	
	Moderate (10–19)	10.8 (6.1)	1.657	(0.720, 2.593)	0.001
	High (20–39)	11.6 (6.2)	2.476	(1.463, 3.488)	<0.0001
ESS	Low (0–9)	7.7 (4.9)	1.000	Reference	
	Moderate (10–19)	7.6 (4.5)	−0.072	(−0.771, 0.626)	0.840
	High (20–39)	8.3 (4.6)	0.514	(−0.240, 1.269)	0.181
BRISC	Low (0–9)	2.1 (1.0)	1.000	Reference	
	Moderate (10–19)	1.9 (0.9)	−0.247	(−0.385, −0.110)	<0.0001
	High (20–39)	1.8 (0.9)	−0.321	(−0.470, −0.172)	<0.0001
PSS	Low (0–9)	25.4 (9.1)	1.000	Reference	
	Moderate (10–19)	28.6 (8.2)	3.273	(2.026, 4.520)	<0.0001
	High (20–39)	30.0 (7.7)	4.760	(3.341, 6.107)	<0.0001
Neighborhood Disorder	Low (0–9)	−0.3 (0.4)	1.000	Reference	
	Moderate (10–19)	−0.2 (0.5)	0.156	(0.083, 0.230)	<0.0001
	High (20–39)	−0.1 (0.6)	0.185	(0.106, 0.264)	<0.0001

## Data Availability

The data are contained within the article.

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
