# Peer review of "Development and Initial Validation of the Assessment of Sleep Environment (ASE): Describing and Quantifying the Impact of Subjective Environmental Factors on Sleep"

_ijerph, 2022, doi:10.3390/ijerph192013599_

Round 1

Reviewer 1 Report

In the present study, the authors developed a 13-item self-report assessment of the sleep environment and tested the reliability and validity of this method. This self-report thoroughly assessed the relationship between environmental factors and various sleep parameters and concluded that certain environmental factors are indeed associated with sleep quality and insomnia symptoms. This is a well-designed study, and rigorous statistical analyses were performed to validate the results. Despite the apparent limitations of self-reports (that are generously admitted by the authors), this assessment does demonstrate reliability regarding relationships between the subjective experience of the sleep environment and different sleep parameters.

I have a few minor comments.

  1. The subjects were aged between 22-60 and included both males and females. Did the authors find any age or sex differences in the results?
  2. Since this is a questionnaire-based study, I guess it will be possible to include subjects of different races, ethnicity, culture, geographical locations, etc., in the future, providing a more comprehensive picture.
  3. To the authors’ knowledge, are there any laboratory-based sleep studies on humans that could validate/support their current findings?

Author Response

COMMENT: The subjects were aged between 22-60 and included both males and females. Did the authors find any age or sex differences in the results?

RESPONSE: This is an interesting question and we have added the following to the Results section: “In univariate analyses, ASE scores did not differ by age (B=0.01, p=0.687) or sex (B=-1.03, p=0.09).”

COMMENT: Since this is a questionnaire-based study, I guess it will be possible to include subjects of different races, ethnicity, culture, geographical locations, etc., in the future, providing a more comprehensive picture.

RESPONSE: We have added the following statement to the Discussion section: “Future studies will be able to explore the ASE in the context of more diverse populations in terms of race/ethnicity, geographic location, nationality, acculturation, amongst others.”

COMMENT: To the authors’ knowledge, are there any laboratory-based sleep studies on humans that could validate/support their current findings?

RESPONSE: To date, the ASE items have not been validated against objective environmental recordings in sleeping areas. We include a statement to this effect in the Limitations section.

Reviewer 2 Report

Dear Authors,

I think it is an excellent study in the development and initial validation of ASE. I think it would be a better study if I added the following points. In the future, I hope that the relationship between objective environmental factors and subjective environmental factors will be clarified.

 Minor improvements

 In Table 1, it is necessary to diagnose the ceiling effect and floor effect by calculating the average value and standard deviation of each item in the 4-point method.

Author Response

COMMENT: In Table 1, it is necessary to diagnose the ceiling effect and floor effect by calculating the average value and standard deviation of each item in the 4-point method.

RESPONSE: We have added the mean and SD for each item in Table 1.